# The Simulation of Vortex Structures Induced by Different Local Vibrations at the Wall in a Flat-Plate Laminar Boundary Layer

**Weidong Cao ***[ID]**, Zhixiang Jia and Qiqi Zhang**

Research Institute of Fluid Engineering Equipment Technology, Jiangsu University, Zhenjiang 212013, China
* Correspondence: cwd@ujs.edu.cn; Tel.: +86-139-5281-6468

**Abstract:** The compact finite difference scheme on non-uniform meshes and the Fourier spectral hybrid method are used to directly simulate the evolution of vortex structures in a laminar boundary layer over a flat plate. To this end, two initial local vibration disturbances, namely, the positive–negative and the negative–positive models, at the wall were adopted. The numerical results show that the maximum amplitudes of vortex structures experience a process of linear growth and nonlinear rapid growth. The vertical disturbance velocity and mean flow shear and the derivative term of the stream-wise disturbance velocity and the span-wise disturbance velocity, are important factors for vortex structure development; the high- and low-speed stripe and the stream-wise vortex are consistent with structures seen in full turbulence. The maximum amplitude of the negative–positive model grows more quickly than that of the negative–positive model, and the detailed vortex structures are different for the two models. The mean flow profiles both become plump, which leads to the instability of the laminar boundary layer. The way in which the disturbance is generated with different local vibrations influences the dynamics of vortex structures in a laminar boundary layer.

**Keywords:** vortex structures; boundary layer; direct numerical simulation; amplitude; vortices

## 1. Introduction

At the beginning of a transition in shear flow, a large number of vortex structures are formed. The formation mechanism and dynamic characteristics of vortex structures are still key issues in fluid mechanics research.

Lee presented direct comparisons of experimental results on transitions in wall-bound flows obtained by flow visualizations, hot-film measurement, and Particle Image Velocimetry (PIV), along with a brief mention of relevant theoretical progresses, based on a critical review of about 120 selected publications. Despite the somewhat different initial disturbance conditions used in the experiments, the flow structures were found to be practically the same [1]. Sharma presented a new theory of coherent structures in wall turbulence. The theory is the first to predict packets of hairpin vortices and other structures in turbulence and their dynamics, based on an analysis of the Navier–Stokes equations, under an assumption of a turbulent mean profile [2]. Wedin studied finite-amplitude coherent structures with a reflection symmetry in the span-wise direction of a parallel boundary layer flow; some states computed displayed a span-wise spacing between streaks of the same length scale as turbulence flow structures observed in experiments [3]. Wu presented a mathematical theory to describe the nonlinear dynamics of coherent structures. The formulation was based on a triple decomposition of the instantaneous flow into a mean field, coherent fluctuations, and small-scale turbulence, but with the mean-flow distortion induced by nonlinear interactions of coherent fluctuations being treated as part of the organized motion [4]. Mcmullan implemented large eddy simulations of the plane mixing layer

for the purpose of reducing the stream-wise vortex structure that may exist in these flows. Both an initially laminar and initially turbulent mixing layer were considered in this study. The initially laminar flow originated from Blasius profiles with a white noise fluctuation environment, whilst the initially turbulent flow had an inflow condition obtained from an inflow turbulence generation method. Flow visualization images demonstrated that both mixing layers contained organized turbulent coherent structures, and that the structures contained rows of stream-wise vortices distributed across the span of the mixing layer [5]. Wall studied three spatially extended traveling wave exact coherent states, together with one span-wise localized state for channel flow. Two of the extended flows were derived by the homotopy method from solutions. Both these flows were asymmetric with respect to the channel center plane, and featured streaky structures in stream-wise velocity flanked by staggered vortical structures; one of these flows featured two streak/vortex systems per span-wise wavelength [6]. Kang studied the direct numerical simulation data of a wave packet in laminar turbulent transition in a Blasius boundary layer. The decomposition of this wave packet into a set of modes could be achieved in a wide variety of ways [7]. Shinneeb experimentally studied the turbulent wake generated by a vertical sharp-edged flat plate suspended in a shallow channel flow with a gap near the bed. Two different gap heights were studied which were compared with the no-gap flow case. The Reynolds number based on the water depth was 45,000. Extensive measurements of the flow field in the vertical and horizontal planes were made using a PIV system. The large vortices were exposed by analyzing the PIV velocity fields using the proper orthogonal decomposition method [8]. Lemarechal experimentally investigated the laminar turbulent transition of a Blasius boundary layer-like flow at the Institute of Aerodynamics and Gas Dynamics, University of Stuttgart. The late stage of controlled transition with K-type breakdown was investigated with the temperature-sensitive paint (TSP) method on the flat-plate surface. The test conditions enabled the TSP method to resolve the complete transition process temporally and spatially. Therefore, it was possible to detect the coherent structures occurring in the late stage of laminar–turbulent transition from the visualizations on the flat-plate surface, namely, $\Lambda$ and $\Omega$ vortices [9]. The effects of isolated, cylindrical roughness elements on laminar–turbulent transition in a flat-plate boundary layer were investigated in a laminar water channel. Most predictions by global linear stability theory could be confirmed, but additional observations in the physical flow demonstrated that not all features could be captured adequately by global linear stability theory [10].

There are many kinds of small disturbance models for boundary layer transition, such as combination Tollmier–Schlichting(T–S) waves, multi-eddy structures which satisfy continuity equation, slot jets, and so on. The initial disturbance sources of other physical models except slot jets are still not clear. In this paper, vortex structures induced by local instantaneous small wall vibration combinations similar to the real physical mechanism are adopted, and the internal mechanisms of the influence of different disturbance combinations on the flow stability are analyzed.

To perform approximate simulations of the wall local forced vibration, an initial disturbance was set in the form of local single-period micro-vibration at the bottom of the wall of $y = 0$. However, the influence of physical deformation on the mesh was neglected since it is outside of the scope of this paper. Two disturbance models were implemented, namely, the positive–negative (P–N) model and the negative–positive (N–P) model. The P–N model is defined as follows: the normal disturbance velocity at mesh points in the circle region, $\sqrt{(x - 10.5)^2 + z^2} < 2$ is supposed to be $v\prime = 0.02 \sin(2\pi \cdot t/15)$, where $0 \leq t \leq 15$, and that in the circle region, $\sqrt{(x - 14.7)^2 + z^2} < 2$, is supposed to be $v\prime = -0.02 \sin(2\pi \cdot t/15)$, where $0 \leq t \leq 15$. Similarly, the N–P model is defined as follows: the normal disturbance velocity at mesh points in the circle region, $\sqrt{(x - 10.5)^2 + z^2} < 2$, is supposed to be $v\prime = -0.02 \sin(2\pi \cdot t/15)$, where $0 \leq t \leq 15$, and that in the circle region, $\sqrt{(x - 14.7)^2 + z^2} < 2$, is supposed to be $v\prime = 0.02 \sin(2\pi \cdot t/15)$, where $0 \leq t \leq 15$. The amplitude of the small disturbance velocity is commonly chosen to be about 1% of the maximum mean basic flow velocity. A relatively small amplitude of the disturbance velocity

can lead to a relatively long time of disturbance evolution. The computational domain and initial disturbance location are shown in Figure 1.

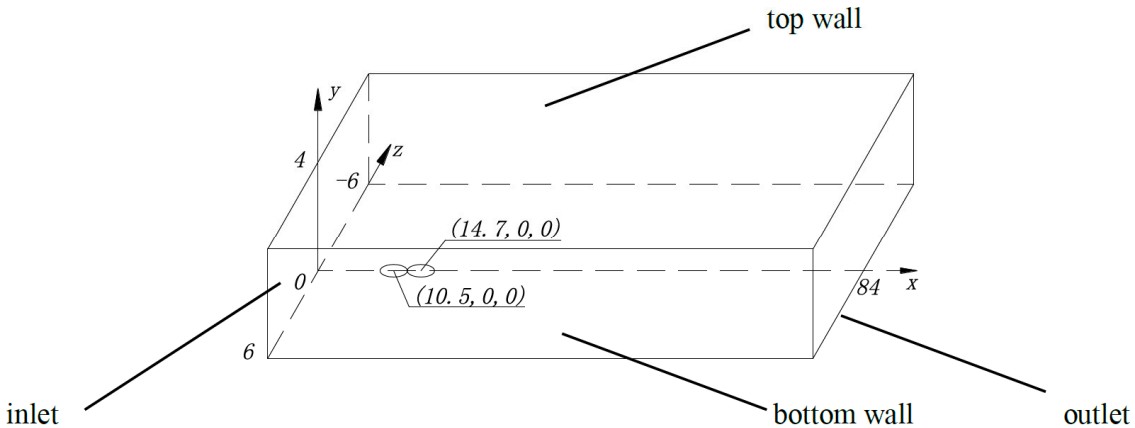

**Figure 1.** Distribution of initial disturbance at the wall.

## 2. Governing Equations and Numerical Method

### 2.1. Governing Equations

The governing equations are the non-dimensional incompressible Navier–Stokes equations and the continuity equation.

$$\frac{\partial \vec{u}\prime}{\partial t} + (\vec{u}_0 \cdot \nabla)\vec{u}\prime + (\vec{u}\prime \cdot \nabla)\vec{u}_0 + (\vec{u}\prime \cdot \nabla)\vec{u}\prime = -\nabla p\prime + \frac{1}{Re}\nabla^2 \vec{u}\prime \tag{1}$$

$$\nabla \cdot \vec{u}\prime = 0, \tag{2}$$

where $\nabla$ is the gradient operator, $\nabla^2$ is the Laplacian, $R_e$ is the Reynolds number, $\vec{u}_0 = (u_0, v_0)$ and $p_0$ are the velocity and the pressure of Blasius solutions, and $\vec{u}\prime = (u\prime, v\prime, w\prime)$ and $p'$ are the disturbance velocity and pressure of vortex structures. In this paper, $R_e = U_\infty \cdot \delta / v = 2000$, where $U_\infty$ is the free stream velocity of Blasius basic flow, $\delta$ is the upstream thickness of the boundary layer in Blasius basic flow, and $v$ is the kinematics viscosity. The velocity of Blasius basic flow $\vec{u}_0 = (u_0, v_0)$ can be obtained through the Falkner–Skan equation, $f''' + f f'' = 0$.

### 2.2. Numerical Methods

The direct numerical simulations of Equations (1) and (2) were implemented as follows: a third-order mixed explicit-implicit scheme was employed for the time discretization, and the space discretization combined the higher-accuracy compact finite differences of non-uniform meshes with the Fourier spectral expansion. The nonlinear terms were approximated by a fifth-order upwind compact difference scheme for non-uniform meshes. The treat of pressure terms was approximated by a third-order center finite difference scheme with five points. The viscous terms were approximated by a fifth-order compact difference scheme for non-uniform meshes. Detailed numerical methods and verifications of simulation accuracy were given in Reference [11].

The computational time step was 0.01. Owing to the limitation of computational capacity, the range of directions $x$, $y$, $z$ was limited to 84, 4, and 12, respectively. The number of Fourier modes was 16, which implies that the number of collocation points in the $z$-direction was 32, and the numbers of mesh points in the $x$- and $y$-directions were 240 and 150, respectively. Uniform and non-uniform meshes were applied in the $x$- and $y$-directions, respectively. The node coordinate $y(k)$ in the $y$-direction can be expressed by Equation (3) below, which is used to refine the mesh in the near-wall region.

$$y(k) = 4[1 - \tanh\tfrac{300-2k}{149}]/\tanh 2 - c_y(150-k)/150$$
$$c_y = 4(1 - \tanh\tfrac{300}{149})/\tanh 2$$

(3)

*2.3. Boundary Layer*

Boundary layer conditions were as follows:

Inflow boundary conditions, $x = 0$, $\vec{u}' = 0$, and $\partial p'/\partial x = 0$;
Outflow boundary conditions, $x = 84$, non-reflecting boundary condition, $\partial p'/\partial x = 0$;
Boundary conditions at the top wall, $y = 4$, $\partial \vec{u}'/\partial y = 0$, $p' = 0$;
Boundary condition at the bottom wall, $y = 0$, $\partial p'/\partial y = 0$, if $0 \le t \le 15$, $v'$ is shown above, $t > 15$, $\vec{u}' = 0$.

To verify the independence of results on mesh and time-step sizes, the numbers of mesh points in the $x$- and $y$-directions were raised to 480 and 200, respectively, and the computational time step was reduced to 0.005. The maximum absolute values of the velocity of the vortex structures $u'$, $v'$, and $w'$ with two kinds of meshes and time steps were compared, as shown in Figure 2. Lines represent the simulation results of the grid and time step used in this paper, and circles represent the simulation results of the raised grid and reduced time step, $t < 25$; there is almost no difference in the comparison of simulation results. Because the simulation efficiency was too low when the number of grids was increased and the time step was reduced, a total of more than one million grids were used in this paper, and the time step was set as 0.01.

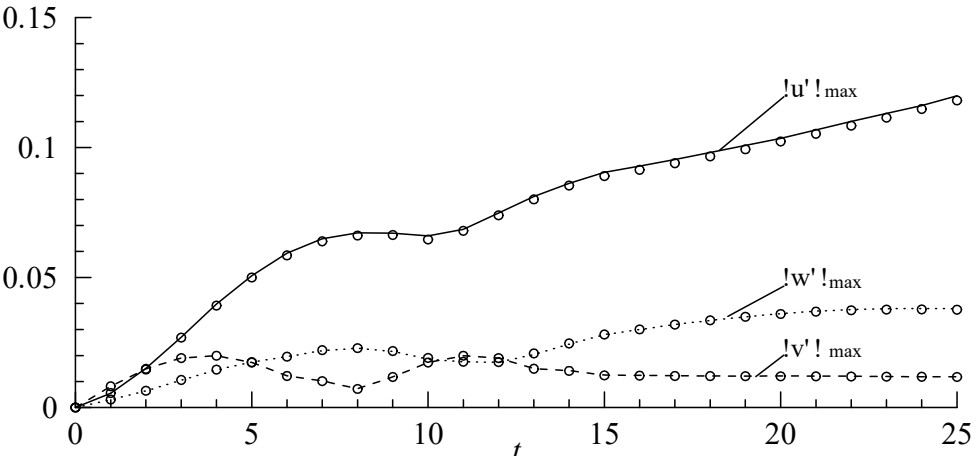

**Figure 2.** Mesh and time-step independence verification.

## 3. Numerical Results and Analyses

*3.1. Disturbance Amplitude of Vortex Structures*

Figure 2 shows the evolution of the maximum disturbance amplitude ($A$) of the vortex structures as they originated from the P–N model and N–P model and propagated downstream in the boundary layer. The maximum disturbance amplitude ($A$) is defined as

$$A = \sqrt{|u'|^2_{\max} + |v'|^2_{\max} + |w'|^2_{\max}}.$$

(4)

At $t = 15$, when the wall local forced vibrations stop, the maximum amplitude of vortex structures derived by the P–N and N–P models was almost 0.1, which is obviously much greater than the initial forced vibration amplitude of 0.02. For $t < 65$, the maximum amplitude of vortex structures derived by

the N–P model showed almost no difference compared to that of the P–N model; in this range, the maximum amplitude of the two models gradually rose from 0 to about 0.35. The maximum amplitude of the N–P model was almost unchanged from $t = 65$ to $t = 80$, and showed rapid growth for $t > 80$. On the other hand, however, the maximum amplitude of $v$ using the N–P model obviously showed a rapid growth for $t > 65$.

It can also be seen from Figure 3 that the largest contribution to the maximum amplitude was that of $|u\prime|_{max}$, with $u\prime$ being the stream-wise velocity, while the second and third contributions were those of $|w\prime|_{max}$ and $|v\prime|_{max}$, respectively. In addition, as $|u\prime|_{max} > 0.6$, both $|w\prime|_{max}$ and $|v\prime|_{max}$ increased gradually.

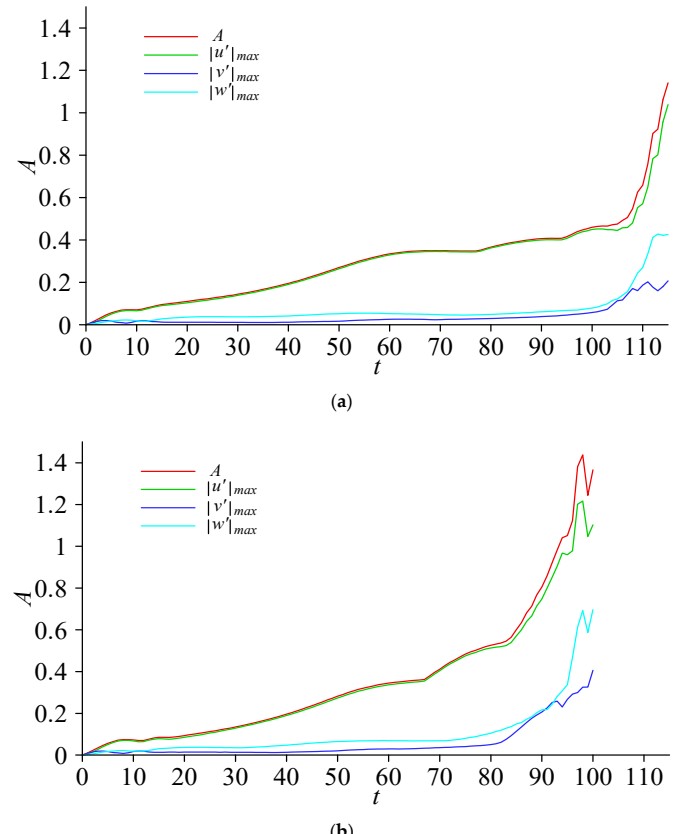

**Figure 3.** Maximum disturbance amplitude: (**a**) positive–negative (P–N) model; (**b**) negative–positive (N–P) model.

In order to study the reasons for the rapid growth of velocity disturbance in the stream-wise direction, the governing equation for the velocity and pressure disturbances of vortex structures in the $x$-direction was written in the following form:

$$
\begin{aligned}
\frac{\partial u\prime}{\partial t} &= \left(-\frac{\partial p\prime}{\partial x}\right) + \left(-u\prime\frac{\partial u\prime}{\partial x} - U_0\frac{\partial u\prime}{\partial x} - u\prime\frac{\partial U_0}{\partial x}\right) + \left(-v\prime\frac{\partial u\prime}{\partial y} - V_0\frac{\partial u\prime}{\partial y} - v\prime\frac{\partial U_0}{\partial y}\right) \\
&+ \left(-w\prime\frac{\partial u\prime}{\partial z}\right) + \frac{1}{\mathrm{Re}}\left(\frac{\partial^2 u\prime}{\partial x^2} + \frac{\partial^2 u\prime}{\partial y^2} + \frac{\partial^2 u\prime}{\partial z^2}\right)
\end{aligned}
\tag{5}
$$

where $u\prime$, $v\prime$, $w\prime$ are the components of the velocity of the vortex structures, $p\prime$ is the pressure disturbance, and $u_0$ and $v_0$ are the stream-wise and vertical velocity components of the Blasius solution. The various terms appearing in Equation (5) can be grouped in the following form, where $b$, $c$, and $d$ are all nonlinear terms, $c = c1 + c2 + c3$, and $e$ is the viscous term:

$$\begin{cases} a = u\prime \times 0.1 \\ b = (-u\prime \frac{\partial u\prime}{\partial x} - u_0 \frac{\partial u\prime}{\partial x} - u\prime \frac{\partial u_0}{\partial x}) \\ c = (-v\prime \frac{\partial u\prime}{\partial y} - v_0 \frac{\partial u\prime}{\partial y} - v\prime \frac{\partial u_0}{\partial y}) \\ c1 = -v\prime \frac{\partial u\prime}{\partial y}, c2 = -v_0 \frac{\partial u\prime}{\partial y}, c3 = -v\prime \frac{\partial u_0}{\partial y} \\ d = (-w\prime \frac{\partial u\prime}{\partial z}) \\ e = \frac{1}{Re}(\frac{\partial^2 u\prime}{\partial x^2} + \frac{\partial^2 u\prime}{\partial y^2} + \frac{\partial^2 u\prime}{\partial z^2}) \end{cases} .$$ (6)

The evolution of grouped terms in time is shown in Figure 4. In particular, Figure 4a shows the evolution trends of $a$, $b$, $c$, $d$, and $e$ at the position of maximum stream-wise velocity disturbance with $u\prime > 0$ for the P–N model; the characteristics of $a$ were similar to those shown in Figure 3a. Compared with the viscous term $e$, the nonlinear terms including $b$, $c$, and $d$ contributed to the acceleration of the growth of $u\prime$, while term $b$ contributed a little. For $t < 55$, it can be seen that $c$ was the most significant term for the disturbance amplitude growth.

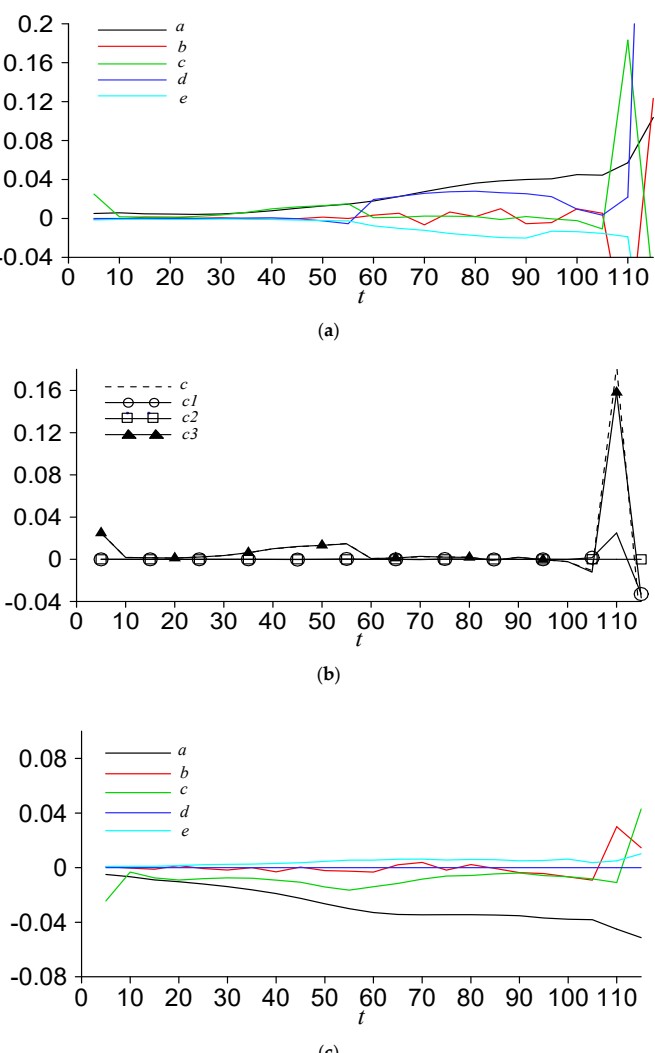

**Figure 4.** *Cont.*

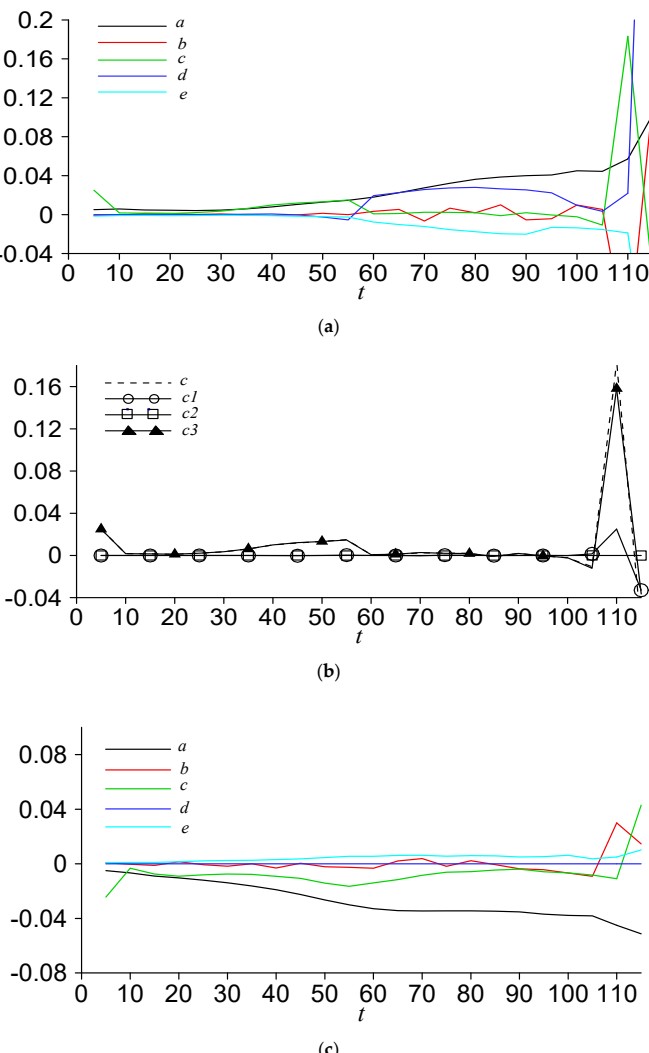

**Figure 4.** Evolution in time of the terms in the governing equations: (**a**) $u\prime > 0$, P–N model; (**b**) $u\prime > 0$, P–N model; (**c**) $u\prime < 0$, P–N model; (**d**) $u\prime < 0$, P–N model; (**e**) $u\prime > 0$, N–P model; (**f**) $u\prime < 0$, N–P model.

Figure 4b shows the evolution trends of $c$, $c1$, $c2$, and $c3$, whereby $c$ was almost equal to $c3$, as $c3$ is the product of $-v\prime$ and the mean shear rate of boundary layer $\partial u_0/\partial y$. Since $\partial u_0/\partial y > 0$, vertical velocity disturbance at the position of maximum stream-wise velocity disturbance was of course negative. However, for $t > 55$, term $d$ becomes important for the disturbance amplitude growth; the span-wise disturbance velocity $w\prime$ and the derivative term of the stream-wise disturbance velocity $\partial u\prime/\partial z$ increase significantly.

Figure 4c shows the evolution trends of $a$, $b$, $c$, $d$, and $e$ at the position of maximum stream-wise disturbance velocity with $u\prime < 0$ for the P–N model. $|u\prime|_{\max}$ was less than that in Figure 3a, which means that the strength of the stream-wise high-speed disturbance velocity was greater than that of the stream-wise low-speed disturbance velocity. Nonlinear terms including $b$, $c$, and $d$ can promote further reduction of the stream-wise disturbance velocity, while viscous term $e$ can prevent it. It can be seen that $c$ was the most significant term for the disturbance amplitude growth, while term $d$ seemed unimportant for $|u\prime|_{\max}$ with $u\prime < 0$. Figure 4d shows the evolution trends of $c$, $c1$, $c2$, and $c3$, whereby $c3$ was almost equal to $c$, and, due to $c3 < 0$, vertical disturbance velocity at the position of maximum stream-wise disturbance velocity was of course positive.

Figure 4e shows the evolution trends of $a$, $b$, $c$, $d$, and $e$ at the position of maximum stream-wise disturbance velocity with $u\prime > 0$ for the N–P model. For $t < 50$, it can be seen that $c$ was the most

significant term for the disturbance amplitude growth. Although |$e$| in Figure 4e was greater than that in Figure 4a, term $d$ in Figure 4e rose very fast for $t > 50$, which was the main reason for the growth of the vortex structures in the N–P model. It can be inferred that terms $w\prime$ and $\partial u\prime / \partial z$ of the vortex structures of the N–P model were relatively greater than those of the P–N model.

Compared to Figure 4c, the stream-wise low-speed disturbance velocity strength of the N–P model shown in Figure 4f was greater than that of the P–N model. Term $c$ was the key factor for the growth of low-speed disturbance velocity. In contrast, terms $c$ and $d$ were the only important factors for the growth of high-speed disturbance velocity.

### 3.2. Stripe of Vortex Structures

Figure 5a provides the velocity vector of the vortex structures for the P–N model at $t = 15$, when the local micro-vibration at the wall stops. Because the disturbance at the wall was symmetrical along the plane $z = 0$, only the results of $0 < z < 6$ are shown. It can be seen that the initial form of the vortex structures was complex, whereby the vortex structures mainly concentrated near $z = 0$, $x = 15$, and showed complex three-dimensional (3D) vortices in space. The core region of disturbance velocity was at the position of $x > 15$ due to the influence of basic flow in the laminar boundary layer.

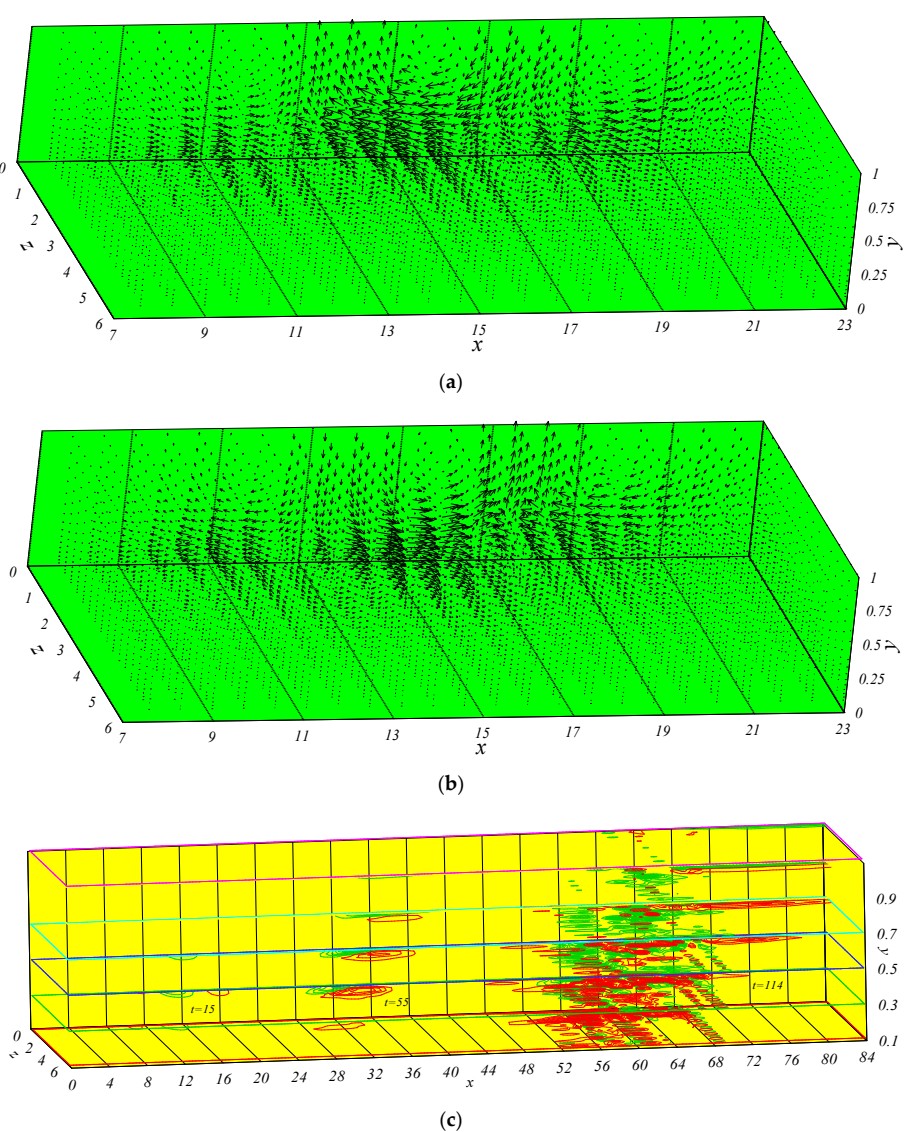

**Figure 5.** *Cont.*

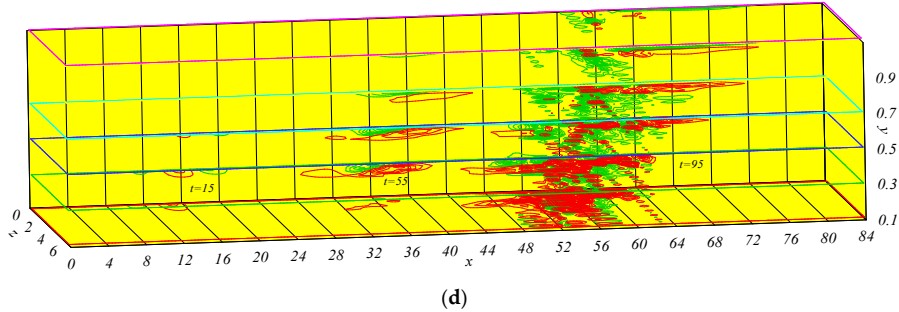

(**d**)

**Figure 5.** Vectors and contours of vortex structures: (**a**) velocity vectors at $t$ = 15, P–N model; (**b**) velocity vectors at $t$ = 15, N–P model; (**c**) stream-wise high- and low-speed disturbance velocity, contour increment of 0.03, P–N model; (**d**) stream-wise high- and low-speed disturbance velocity, contour increment of 0.03, N–P model.

Figure 5b provides the velocity vector of the vortex structures at $t$ = 15 of the N–P model. The contrast between Figure 4a,b shows the opposite nature of the vortices; however, in fact, there were differences in the amplitudes of the velocities. For example, at the position of $x$ = 18.2, $y$ = 0.309, $z$ = 0, the velocities of the P–N model and N–P model are compared in Table 1.

**Table 1.** Velocity comparison. P–N—positive–negative; N–P—negative–positive.

| Model | $u^{'}$ | $v^{'}$ | $w^{'}$ |
|---|---|---|---|
| P–N | $2.260294610168348 \times 10^{-2}$ | $-5.310678689829363 \times 10^{-3}$ | $5.266260498348791 \times 10^{-18}$ |
| N–P | $-6.111656282953886 \times 10^{-2}$ | $7.780919736954207 \times 10^{-3}$ | $-2.068746728425687 \times 10^{-18}$ |

Apart from the different spatial evolution in time, the stream-wise high-speed and low-speed disturbance velocities of different vortex structures exhibited different properties. The vortex structure amplitude of the P–N model increased to about 1.0 at $t$ = 114, and that of the N–P model increased to about 1.0 at $t$ = 95. Figure 5c provides the stream-wise disturbance velocity distributions of high-speed and low-speed fluids of the P–N model at $t$ = 15, 55, and 114, at $y$ = 0.1, 0.3, 0.5, 0.7, 0.9, and 1.1. Because the vortex disturbances were symmetrically distributed along plane $z$ = 0, only the contour between planes of $z$ = 0 and $z$ = 6 is displayed; the contour increment is 0.03, and red lines represent the disturbance velocity of the high-speed fluid, $u\prime$ > 0. Green lines represent the disturbance velocity of the low-speed fluid, $u\prime$ < 0. At $t$ = 15, high-speed and low-speed fluids mainly distributed in a local area near $x$ =16–20, $y$ = 0.3, $z$ = 0, and the intensity of the low-speed fluid was greater than that of the high-speed fluid, with the former being below the latter. With the evolution of the vortex structure, at $t$ = 55, the intensity and area of both the high-speed and low-speed fluids increased, but it seems that the area of the high-speed fluid was larger than that of the low-speed fluid. The low-speed fluid concentrated near the plane of $z$ = 0, and the high-speed fluid existed in relatively large areas. At $t$ = 144, the vortex structure was further inclined due to the shear action of the basic flow in the laminar boundary layer. At the same time, it can be seen that the high-speed fluid mainly concentrated near the wall, which may have led to an increase in the friction shear force at the wall region. The area occupied by the high-speed fluid was larger than that occupied by the low-speed fluid. The spatial range of the high-speed fluid increased more than that of the low-speed fluid in all directions.

Figure 5d provides the stream-wise disturbance velocity distributions of the high-speed and low-speed fluids of the N–P model at $t$ = 15, 55, and 114, at $y$ = 0.1, 0.3, 0.5, 0.7, 0.9, and 1.1. Compared with Figure 4c, at $t$ = 15, although the high-speed and low-speed fluids mainly distributed in a small area near $x$ = 16–20, $y$ = 0.3, $z$ = 0, the intensity of the high-speed flow was slightly greater than that of the low-speed flow, while the high-speed fluid was lower than the low-speed fluid. The low-speed fluid main distributed at the location near $y$ = 0.5 and $x$ = 20, and there was no high-speed fluid at

the same position in Figure 5c. At *t* = 55, the high-speed fluid exceeded the low-speed fluid, and the elongation range of the high-speed fluid in the stream-wise direction was larger than that in Figure 4c. At *t* = 95, the amplitude of vortex structures rose rapidly to nearly 1.0, and the scales of the high-speed fluid and low-speed fluid were smaller than those in Figure 5c in the stream-wise direction; however, there were mainly high-speed stripes near the wall, and the characteristics of the high-speed fluid occupied a larger area than the low-speed fluid, similar to Figure 5c. Results of the vortex structure stripe agree with the results of Wall et al. [12].

The rough center positions of the high-speed and low-speed fluids in the plane of *y* = 0.3 of the P–N and N–P models are listed in Table 2. The forward speed in the stream-wise direction of the P–N model was approximately $0.45U_\infty$, and that of the N–P model was approximately $0.525U_\infty$.

**Table 2.** Center position evolution with time.

| Model | | *x* | |
|---|---|---|---|
| P–N | 16 (*t* = 15) | 34 (*t* = 55) | 60 (*t* = 114) |
| N–P | 16 (*t* = 15) | 38 (*t* = 55) | 58 (*t* = 95) |

### 3.3. Mean Flow Profile and Neutral Curve

The black dashed lines in Figure 6a,b represent the velocity $u_0$ of Blasius basic flow at *x* = 64 for the P–N model and at *x* = 56 for the N–P model. The red solid lines in Figure 6a,b represent the mean value of the stream-wise disturbance velocity within a local region adding to Blasius basic flow; the local region was 52 < *x* < 76, −6 < *z* < 6 for the P–N model and 44 < *x* < 68, −6 < *z* < 6 for the N–P model. As can be seen, the mean velocity profile deformations in the area of the vortex structures could easily be discerned. On one hand, because of the presence of stream-wise disturbance velocity, the shear stress close to the wall increased. On the other hand, under the conditions of different initial disturbances at the walls in the laminar boundary flow, the mean velocity profiles all became plump after a period of evolution. Although vortex structures were only in their initial stages, the velocity profiles had a tendency to evolve into the turbulent mean velocity profiles. Furthermore, the mean velocity profiles were plumper at *y* < 0.3, which indicates that the stream-wise high-speed disturbance velocity mainly distributed near the wall region.

The green solid lines in Figure 6a,b represent the mean value of the stream-wise disturbance velocity within a local scope adding to Blasius basic flow; the local region was 52 < *x* < 76, −3 < *z* < 3 for the P–N model and 44 < *x* < 68, −3 < *z* < 3 for the N–P model. The black solid lines in Figure 6a,b represent the mean value of the stream-wise disturbance velocity within a local region adding to Blasius basic flow; the local region was 52 < *x* < 76, −1.5 < *z* < 1.5 for the P–N model and 44 < *x* < 68, −1.5 < *z* < 1.5 for the N–P model. It can be seen from Figure 6a,b that the mean velocity profiles had inflection points in the local scope of −1.5 < *z* < 1.5; the flow stability in this scope would, thus, be altered. According to the theory of linear stability, the solution of a three-dimensional T–S wave is obtained by solving the eigenvalue problem of Orr–Sommerfeld equations.

$$\vec{u}\prime = a_0\left[\vec{u}(y)\right]e^{i(\alpha x+\beta z-\omega t)} + c.c,\tag{7}$$

where $c \cdot c$ is the conjugate complex, $\alpha = \alpha_r + i\alpha_i$ is the stream-wise wave number, $\alpha_r$ is the real part, $\alpha_i$ is the imaginary part, $\beta$ is the span-wise wave number, $\omega$ is the frequency, $a_0$ is the initial amplitude, $\vec{u}(y) = \{u(y), v(y), w(y)\}$ is the eigenvalue velocity, and $\alpha_i = 0$ represents the points on the neutral curve. For each span-wise wave number $\beta$, the corresponding unstable T–S wave with maximum local growth rate $-\alpha_i$ exists.

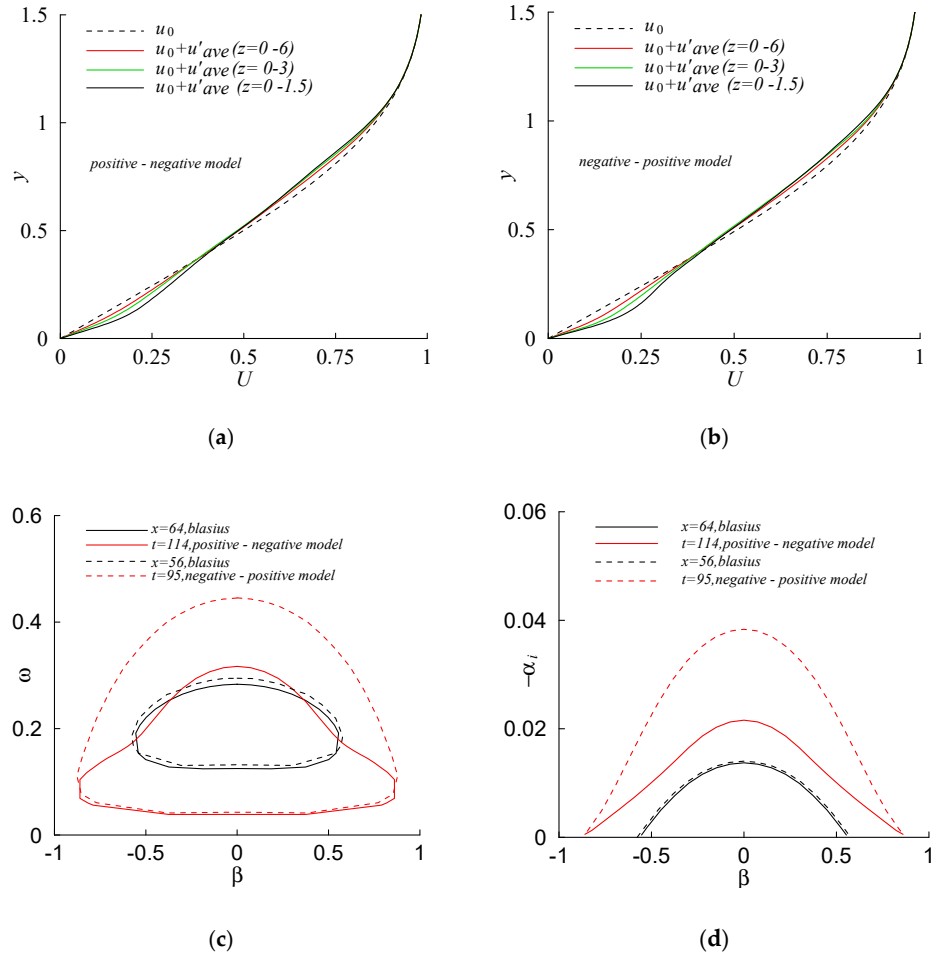

**Figure 6.** Mean stream-wise velocity and flow stability: (**a**) $t = 114$; (**b**) $t = 95$; (**c**) neutral curve; (**d**) growth rate.

T–S waves inside the neutral curve are unstable. The black solid line in Figure 6c is the neutral curve at $x = 64$, and the black dotted line is the neutral curve at $x = 56$ in the Blasius boundary layer. The frequency of the neutral curve at $x = 64$ was less than that at $x = 56$. The red solid line in Figure 6c is the neutral curve based on the mean value of the stream-wise disturbance velocity added to the Blasius basic flow within the local region $52 < x < 76$, $-1.5 < z < 1.5$ of the P–N model. The range of the neutral curve (red solid line) was obviously larger than that of the black solid line. The red dashed line in Figure 6c is the neutral curve based on the mean value of the stream-wise disturbance velocity added to the Blasius basic flow within the local region $44 < x < 68$, $-1.5 < z < 1.5$ of the N–P model. The range of the neutral curve (red dashed line) was obviously larger than that of the black dashed line. Due to the existence of the vortex structures, the neutral curve range of the N–P mode was relatively the largest.

The black solid and dashed lines in Figure 6d correspond to the maximum local growth rates $-\alpha_i$ at $x = 64$ and $x = 56$ in the Blasius boundary layer. The red solid line in Figure 6d is the maximum local growth rate $-\alpha_i$ based on the mean value of the stream-wise disturbance velocity added to the Blasius basic flow within the local region $52 < x < 76$, $-1.5 < z < 1.5$ of the P–N model. The amplitude of the maximum local growth rate $-\alpha_i$ of the red solid line was obviously larger than that of the black solid line. The red dashed line in Figure 6d is the maximum local growth rate $-\alpha_i$ based on the mean value of the stream-wise disturbance velocity added to the Blasius basic flow within the local region $44 < x < 68$, $-1.5 < z < 1.5$ of the N–P model. Also, the amplitude of the maximum local growth rate $-\alpha_i$ of the red dashed line was larger than that of the black dashed line. The amplitude of the maximum local

growth rate $-\alpha_i$ of the N–P model was relatively the largest. Growth rates of the T–S waves and profile characteristics of the mean flow had the ability to promote each other. The self-sustaining structures in the logarithmic region of the boundary agree with the results obtained by Yang [11].

### 3.4. Stream-Wise Vortices

Figure 7a shows the distribution of stream-wise vortices $\omega_x$ at different times of $t$ = 15, 55, and 114 of the P–N model. Figure 7b shows the distribution of stream-wise vortices $\omega_x$ at different times of $t$ = 15, 55, and 114 of the N–P model. Contours in three section planes at each time are given, and the contour increment was 0.1; the red color denotes $\omega_x > 0$, while the green color denotes $\omega_x < 0$.

$$\omega_x = \partial w\prime/\partial y - \partial v\prime/\partial z. \tag{8}$$

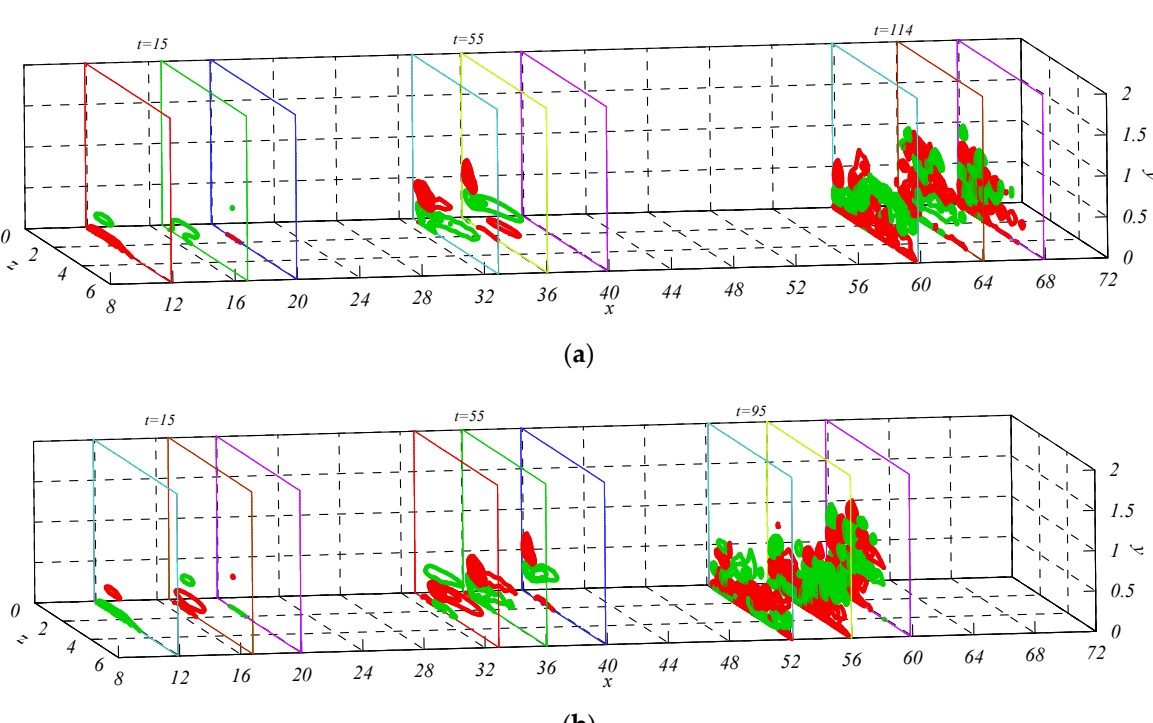

**Figure 7.** Stream-wise vortices: (**a**) P–N model; (**b**) N–P model.

The intensity of stream-wise vortices became stronger and stronger with the evolution of vortex structures. Furthermore, different scales of stream-wise vortices came into being, the centers of vortices went up, and the influencing areas gradually diffused, followed by strong ejection and sweeping at the centers of these vortices. In addition, the sizes of stream-wise vortices increased continuously. According to Biot–Savart theory, the vortex tube is lengthened in the stream-wise direction, which amplifies the vortex intensity. At last, both stream-wise and span-wise velocities grow quickly, followed by dramatic ejection and sweeping, the formation of strong shear layers locally, and the flow becoming unstable. However, the main affected area is in the vicinity of the wall, which is the region near the wall where hydrodynamic instability occurs actively. Results of the vortex distribution agree with the research results of Shinneeb et al. [12].

## 4. Conclusions

A direct numerical simulation of the nonlinear evolution of vortex structures in a laminar boundary layer induced by different local micro-vibrations on the wall was carried out. The characteristics of vortex structures were studied in detail, and the conclusions are summarized below.

With the evolution of time, the amplitude of vortex structures undergoes a process of linear increase and nonlinear rapid increase, and the amplitude of the stream-wise velocity disturbance is the relative maximum. The growth rate of vortex structures in the P–N model is slightly less than that in the N–P model, and the growth of the P–N model vortex structure is most affected by the vertical velocity and boundary layer shear. The growth of vortex structures in the N–P model is greatly influenced by the derivative term of the stream-wise disturbance velocity and span-wise disturbance velocity. In the core region of vortex structures, the inflection point of the velocity profile exists, which leads to the expansion of the range of neutral curves, as well as the increased growth rate of T–S waves. The change in mean flow profile further induces or promotes the growth or formation of vortex structures. These results will help to understand and further study how coherent structures in turbulence come into being and evolve.

**Author Contributions:** W.C. designed the scheme and wrote the program, organized paper; Z.J. helped to analyze the data; Q.Z. contributed some figures.

**Funding:** This work was supported by the National Key R&D Program of China (2018YFC0810506) and the Key R&D Program of Zhenjiang (SH2017049, BK20161472).

**Conflicts of Interest:** The author(s) declared no potential conflicts of interest with respect to the research, authorship, and/or publication of this article.

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
