# Peer review of "The Simulation of Vortex Structures Induced by Different Local Vibrations at the Wall in a Flat-Plate Laminar Boundary Layer"

_processes, doi:10.3390/pr7090563_

Round 1

Reviewer 1 Report

The authors investigate an interesting topic, that of the formation and evolution of vortex structures in a laminar boundary layer beginning from a disturbance in the flowfield. Their approach is based on numerical simulations in cases that are generally well organized. However:

- The use of the English language is too poor. Due to this fact, this reviewer was not able to critically read the text. In his effort to make the text understandable, he made and proposes to the authors some a lot of modifications to the text.The authors are not oblidged to follow them. They could (or should) give the text to a language professional in order to correct it and make it understandable.

- Although the literature review (section 1 - Introduction) is complete enough, it is tiring to the reader. In my opinion, it should be organized in paragraphs referring to works of similar topic. In addition, a larger and better paragraph should be written to describe what is presented in the paper, after having presented the relevant literature.

- The numerical methods (section 2.2) should be enriched with details some on the numerical scheme, as well as appropriate references.

- The test case in not described well or it is not described at all!. What is the Reynolds number of the base flow? The inlet velocity? The reader should be able to easily find the base case, e.g. incompressible flow over a flat plate, where disturbances are produced in order ... The authors have described the model they used to initiate the disturbances in the field, but they have not referred anything about the base field with the Blasius solution. After adding a section with the above, consider also renaming section 2.3 to something like 'Disturbance models'. In In addition, the boundary conditions are included below the legend of Fig. 1 in small fonts, not linked to anything. I think they should be described normally in the text, after having defined the case well.

- The presentation of results (section 3 and its subsections) is too long and "tiring" for the reader. The authors make a lot of descriptions about the bahavior of the velocity disturbance components, but they should find a way to present and discuss them in a more compact way, as well as to state partial conclusions, for example after having discussed each figure. For example, in section 3.3, they repeat a lot of times the same way ito locate a region (local scope?) in a (excuse me for this) tiring way, difficult for the reader to follow. They should find and examine other more compact and appropriate ways of presentation (use of tables?), instead of making descriptionsand resulting in a  lengthy text.

I also suggest to the authors to see the attached scanned document. Apoart from the proposed corrections, they will find various remarks and proposals in it.

According to the above, this reviewer judges that the paper requires major revision.

Reviewer 2 Report

The manuscript provides numerical simulations of disturbance evolutions in a laminar boundary layer. The overall methodology is scientifically sound and reasonable. However, the following should be considered before the manuscript is accepted for publication:

1) the authors refer to their numerical method as direct numerical simulation (DNS). However, for the low Reynolds numbers considered in this study, the flow is always laminar. As DNS is a term commonly used for turbulent flow simulations, the reviewer believes that it should not be used here.

2) The significance of this study is obscure. The introduction should be revised in such a way that the reader will understand the gap in the knowledge and the contribution of the current manuscript. The reviewer assumes that the main interest in this topic comes from transition to turbulence which is not covered in the current simulations. What are the laminar flow applications where the results from this study could be useful for?

3) The authors have used a relatively small number of mesh points, i.e. slightly more than 1 million. How did the authors arrive at this number? How different the results will be if different resolutions are used? The dependence of results on mesh and time-step sizes should be clearly discussed in the manuscript.

4) The English language structure needs to be carefully revised. Specifically, the authors have frequently used very long sentences. These long sentences contain multiple statements which should be broken into separate sentences. As an example, refer to the second sentence in the Abstract.

5) Many of the works referenced in this manuscript are more than a decade old. While it would be acceptable to refer to old publications, the reviewer believes that the evolution of the state of the art in recent years are not well covered in the manuscript.

Round 2

Reviewer 1 Report

The work of the authors is interesting enough and of good level for an always "hot" research item.

From this reviewer point of view, the paper is almost ready to be published. Some minor corrections and questions have been written in the form of sticker comments in the attached pdf file. These have to be taken into account by the authors (however, this reviewer does not need to check the paper again).

Author Response

Thank you very much for your letter and the comments about our paper. We have checked the manuscript and revised it again according to the comments. We submit here the revised manuscript as well as a list of changes. Words in red are the changes we have made in the paper, and the verification of independence of mesh and time step has been added, two tables about velocity difference and forward speed have been added too. Other figures are intuitive , changing some figures into tables will lead to plenty of data and difficult to distinguish, so figures have been have been reserved.

Reviewer 2 Report

I have seen that the authors have improved the manuscript based on the comments from both reviewers. However, before being appropriate for publication, the manuscript needs to be further revised.

1) The dependence of results on mesh and time step sizes should be clearly discussed in the manuscript. The authors have referred to Reference [10] in their response to my comments, but it is important that a brief discussion of the mesh and time-step studies be provided in the manuscript for the general readers.

2) The English language still needs significant improvement. I recommend professional editing before sending back for publication.

3) The presentation of results is still poor. I recommend the authors extract the most important findings and present in Tables with brief discussions of the findings. The current presentation of results is too long and mostly qualitative. A more concise and quantitative presentation will make the manuscript a lot more readable.

Author Response

(The authors gave the same response as above.)

Round 3

Reviewer 2 Report

The manuscript is now ready for publication.